# Pathogenesis of Type 1 Diabetes: Established Facts and New Insights

**DOI:** 10.3390/genes13040706

**Published:** 2022-04-16

**Authors:** Ana Zajec, Katarina Trebušak Podkrajšek, Tine Tesovnik, Robert Šket, Barbara Čugalj Kern, Barbara Jenko Bizjan, Darja Šmigoc Schweiger, Tadej Battelino, Jernej Kovač

**Affiliations:** 1Division of Paediatrics, University Medical Centre Ljubljana, 1000 Ljubljana, Slovenia; ana.cirnski@kclj.si (A.Z.); katarina.trebusakpodkrajsek@mf.uni-lj.si (K.T.P.); tine.tesovnik@kclj.si (T.T.); robert.sket@kclj.si (R.Š.); barbara.kern@kclj.si (B.Č.K.); barbara.jenko.bizjan@kclj.si (B.J.B.); darja.smigoc@kclj.si (D.Š.S.); tadej.battelino@mf.uni-lj.si (T.B.); 2Department of Paediatrics, Faculty of Medicine, University of Ljubljana, 1000 Ljubljana, Slovenia

**Keywords:** type 1 diabetes, genetics, β-cell, epigenetics, viral infections

## Abstract

Type 1 diabetes (T1D) is an autoimmune disease characterized by the T-cell-mediated destruction of insulin-producing β-cells in pancreatic islets. It generally occurs in genetically susceptible individuals, and genetics plays a major role in the development of islet autoimmunity. Furthermore, these processes are heterogeneous among individuals; hence, different endotypes have been proposed. In this review, we highlight the interplay between genetic predisposition and other non-genetic factors, such as viral infections, diet, and gut biome, which all potentially contribute to the aetiology of T1D. We also discuss a possible active role for β-cells in initiating the pathological processes. Another component in T1D predisposition is epigenetic influences, which represent a link between genetic susceptibility and environmental factors and may account for some of the disease heterogeneity. Accordingly, a shift towards personalized therapies may improve the treatment results and, therefore, result in better outcomes for individuals in the long-run. There is also a clear need for a better understanding of the preclinical phases of T1D and finding new predictive biomarkers for earlier diagnosis and therapy, with the final goal of reverting or even preventing the development of the disease.

## 1. Introduction

Type 1 diabetes (T1D) is traditionally viewed as a disease of pancreatic β-cells being attacked by autoreactive T lymphocytes, leaving a person in danger of hyperglycaemia- and hypoglycaemia-related complications [1]. Although its prevalence is only up to 10% of the total diabetes cases worldwide, its incidence is rising globally [2].

T-lymphocyte-mediated insulitis, followed by the presence of one or more type of autoantibody (AAb) against insulin, glutamic acid decarboxylase (GAD), protein tyrosine phosphatase IA-2 or IA-2β, and zinc transporter 8 (ZnT8), is indicative of the immunological onset of T1D [1,3,4]. More than one type of AAb present marks stage 1 of T1D. Stage 2 is marked by dysglycaemia or glucose intolerance. Both stages are asymptomatic; stage 3, however, is defined by the clinical presentation with symptoms of hyperglycaemia (polyuria, polydipsia, enuresis, weight loss, blurred vision), sometimes even with diabetic ketoacidosis (DKA) or diabetic hyperosmolar syndrome [1,3]. Individuals with T1D are also more susceptible to other autoimmune diseases, such as Hashimoto’s thyroiditis, coeliac disease, Addison’s disease, vitiligo, and myasthenia gravis [1,5]. Moreover, the association of T1D with thyroid autoimmunity and coeliac disease may also be mechanistically determined [6,7,8].

The long-term complications include microvascular and macrovascular defects, while hypoglycaemic events and their severity largely depend on long-term glycaemic control, reflected in the blood glucose and HbA_1c_ concentrations. Glycaemic control usually correlates with a person’s socioeconomic status and age, and individuals with T1D report overall poorer quality of life due to the psychological stress they experience [8].

It is becoming apparent that T1D is a highly heterogeneous disease influenced by a complex network of different factors, such as age, genetic predisposition, and environmental interactions, and that pancreatic β-cells play a significant role in initiating pathogenic processes through the crosstalk with immune cells [9]. The heterogeneity is apparent at the tissue level (frequency and identity of cellular infiltrates in islets) and at the bedside as individuals differ in the progression of the disease and response to therapies [10].

## 2. Pathology of β-Cells

The early immune responses triggering insulitis are innate and include the activation of pattern recognition receptors by endogenous “danger signals” or exogenous ligands produced during viral infections on β-cells, which is a possible link between environmental risk factors and the development of T1D [11]. This results in type I interferon (such as IFNα) production by the β-cells and other cells present in islets, which initiates the recruitment of immune cells. Macrophages are among the first responders and the main cell type producing TNFs. This triggers NF-κB activation in β-cells, which is mostly pro-apoptotic [12]. In the later stages, the inflammatory microenvironment in the pancreatic islets increases the vascular permeability and facilitates the infiltration of naïve and non-islet-reactive T cells in addition to activated T cells [13]. The infiltrates are of predominantly CD8^+^ T type but also include CD20^+^ B cells, CD4^+^ T cells, and CD68^+^ macrophages [14,15]. They are found within and around pancreatic islets [14]. The β-cell response to cytokines, such as IL1-β and IFNγ, present in this stage is the activation of the anti-inflammatory pathways (i.e., IL10, IL4/13) and immune checkpoint proteins (i.e., PDL-1 and HLA-E) [12,16,17]. Furthermore, pro-inflammatory cytokines disrupt β-cells’ metabolic and electrical activity, insulin granule synthesis and content, and gap junction coupling, as well as excessive generation of reactive oxygen species (ROS) and activation of caspases [18,19]. Both types of immunity remain present throughout the course of the disease [12].

The evidence points to two age-related disease endotypes, namely T1DE1 and T1DE2 (type 1 diabetes endotype 1 and 2, respectively). In T1DE1, individuals are younger at the onset, more frequently carry the *HLA-DR4/DQ8* allele, and initially develop anti-insulin autoantibodies (IAA). The hallmarks of this endotype are insulitis with many CD8^+^ T cells and CD20^high^ B cells, few residual insulin-containing islets (ICIs), and abnormal insulin processing in the remaining β-cells. In T1DE2, individuals are older (>13 years) at the time of clinical manifestation, usually carry the *HLA-DR3/DQ2* allele, and initially present with anti-GAD Abs (GADA). Insulitis with low levels of CD8^+^ T cells, the CD20^low^ B-cell phenotype, and more residual ICIs and normal insulin processing are indicative of this type, with weaker evidence for a major T cell role in the pathogenesis [15,20]. Heterogeneity and possible subtypes of T1D pathophysiology were corroborated by a recent study of pancreas tissue slices from organ donors, which correlates anatomical and physiological insights. Alongside the islet infiltration by immune cells and evidence of β-cell mass reduction, dysfunction of β-cells was observed [21,22]. This could represent a critical early event in T1D pathogenesis as β-cell dysfunction exists years before the clinical presentation of T1D [23].

Indeed, it is becoming obvious that the pathogenesis of T1D involves both pancreatic β-cells and immune cells and that the crosstalk between them is of utmost importance in T1D development [9]. β-cells are not just passive targets but actively participate and possibly amplify pathogenic processes [24,25]. They employ compensatory mechanisms in response to immune stress, which become deleterious in the long-run. During increased insulin synthesis or other environmental stresses, they adapt their ER and mitochondria functions by triggering the unfolded protein response (UPR) to restore cellular homeostasis [26,27,28]. Increased production of proteins can lead to increased concentrations of misfolded proteins and the accumulation of defective ribosomal products, which serve as neoepitopes in the HLA-I presentation pathway. Similarly, other granule proteins, expressed with insulin during the β-cell glucose response, contribute to the HLA-I peptidome presented by β-cells [13,29,30,31,32]. Interestingly, at the same time, down-regulation of the pathways involved in the integration of energy metabolism and the regulation of gene expression in β-cells occurs [26]. The genes involved in those pathways are critical for insulin release and the maintenance of the β-cell phenotype and function [17]. Furthermore, proinflammatory signalling (following viral infection) induces the dedifferentiation of β-cells, characterized by a decreased expression of the key β-cell genes involved in maintaining β-cell identity and function, leading to the loss of insulin production [33,34].

Recent findings also reveal the possible involvement of hybrid and chimeric neoepitopes, formed in the transpeptidation reaction in stressed β-cells, in diabetogenic CD4^+^ T-cell activation [35,36,37,38,39]. In connection to this, HLA-I hyperexpression has been described in recently diagnosed and long-term individuals with T1D, as well as in AAb+ donors before the clinical presentation of the disease. Interestingly, HLA-I was mostly expressed by α-cells regardless of disease status [40]. They were also found to be functionally impaired, expressed proinflammatory signals, and had altered gene expression [41,42,43]. Still, β-cells are more affected by this process, suggesting the immune attack of β-cells is related to the presence or presentation of insulin [40]. Increased expression of HLA-II and the components in the HLA-II antigen processing and presentation pathway was observed in the β-cells of individuals with T1D as well, and it appears to be unique to T1D. This suggests a direct role for β-cells in T1D pathology, acting as professional APCs (antigen-presenting cells) in the autoimmune response [44]. The exposure of human β-cells to pro-inflammatory cytokines IFNα, IL1-β, and IFNγ results in chromatin remodelling, alternative splicing, and first exon usage, leading to the differential expression of genes, most notably increased expression of HLA-I, which, together with ER stress and β-cell apoptosis, may lead to an increased presentation of neoantigens, thus contributing to the recruitment of auto-reactive CD8^+^ T cells that selectively attack β-cells [16,45]. Neoantigens can also be generated via differential mRNA splicing and posttranslational modifications, both part of the β-cell stress response [13,30].

β-cells, as part of the endocrine system, are capable of long-distance communication by the secretion of insulin and other granule proteins, as well as exosome-derived proteins (GAD56 and IA-2) directly into the bloodstream. Under stress, they also secrete high amounts of (pre)proinsulin and insulin peptides [13]. Furthermore, stress, such as elevated cytokine levels, impairs cell–cell communication, which is necessary for normal insulin secretion [18]. Endocrine secretion of autoantigens facilitates their uptake by APCs outside of the pancreas, again emphasising the active role of β-cells in potentiating immune responses against themselves [13]. Figure 1 outlines the interplay of different pathogenic influences initiating the autoimmune response on β-cells during T1D progression to overt disease.

## 3. Genetic Predisposition

The overall risk of T1D in the general population is moderately low (with the exception of some populations, such as Sardinian and Finnish, with over 35% incidence), and most studies found significant differences in the genetic, immunologic, metabolic, and clinical characteristics of T1D in people with different ethnic backgrounds. This risk is 15-fold higher among the relatives of individuals with T1D [46]. Interestingly, children in T1D families have different risks depending on the sex of a person with T1D [47,48]. Another proof for the genetic contribution to T1D is an association between disease-causing variants in the genes connected with immune function and the occurrence of T1D. Studies largely employ two different approaches for studying T1D genetics, namely candidate gene association studies and genome-wide linkage analysis studies (GWAS) [46,49]. Some of the genes connected to T1D pathology are shown in Table 1.

A genetic link between T1D and T2D is of special interest as the associated loci for them were thought to be almost completely separated despite the partially shared phenotype. Newer studies, however, found several shared risk genes or genetic polymorphisms [59,61,62,63,64]. Some of the genes located in shared risk loci interact to mutually regulate important islet functions that are disturbed by disease-associated variants, leading to β-cell dysfunction [61]. The regulatory impacts on shared genes and pathways generate overlapping biological mechanisms, which mediate pleiotropic effects on T1D and T2D. Interestingly, a high-risk genetic profile for T1D modifies the biological pathways that increase the risk of developing both T1D and T2D but not vice versa [65].

The *DR3-DQ2* and *DR4-DQ8* haplotypes of HLA class II genes are well established as major risk factors of T1D [66,67,68], and at least one is present in 90% of individuals with T1D and both in 40% (for apparently healthy controls, these numbers are 30% and 3%, respectively). However, they only partly explain the increased risk for T1D development as they influence seroconversion but not the age of the onset, clinical progression, or C-peptide loss [15]. HLA-I alleles *HLA-A*24* and *HLA-B*39* and non-HLA genes, such as *INS, CTLA4, PTPN22,* and *IL-2RA,* are also widely accepted as having an impact in T1D and could improve the understanding of disease heterogeneity [49,69]. In this domain, at least a part of the missing genetic component can be attributed to undetected rare and low frequency variants in at least 40 different loci, which have larger effect sizes and play more significant roles in susceptibility to common diseases, including T1D [70,71].

Protective alleles have also been described, which are present more frequently in the general population as opposed to individuals with T1D [46,72]. For example, the *DR1501-DQ6* protective haplotype has been linked to higher frequencies of islet Ag-specific CD4^+^ effector T (Teff) and regulatory T (Treg) cells. Interestingly, Teffs, although still capable of expressing IFN-γ and IL-4, were more likely to express IL-10 when compared to Teffs from people with susceptible haplotypes, and Tregs successfully suppressed Teffs in an Ag-specific manner, indicating a possible mechanism of action for a haplotype-dependent effect on T1D pathology [73]. Since T1D and coeliac disease share their main susceptibility alleles, *HLA-DQ2* and *HLA-DQ8*, which contribute to the coexistence of both diseases, it is plausible that some other genes also have an influence. Research on the Slovenian population revealed a dual role of *PTPN22* rs2476601 polymorphism in increased risk for T1D and protection against coeliac disease [74].

The genetic pathways involved in different stages of the disease are highly divergent, and genetics is currently the only tool capable of detecting those at risk before the development of islet autoimmunity [75]. More and more genetic-based evidence points to the predisposition at the β-cell level, with risk variants affecting the susceptibility to pro-apoptotic stimuli and influencing β-cell phenotypes [50,57,76]. New in-depth analyses of the potential role of genetic variants in the progression from islet autoimmunity to clinical T1D at the genome-wide level using next-generation sequencing identified several novel risk genes associated with T1D. They are associated with the progression from islet autoimmunity to overt T1D and are particularly involved in the pathways critical to the response to viral infections and interferon signalling in β-cells and autoimmunity development in T cells [55,66]. In fact, more than 80% of the identified candidate genes in T1D are expressed in β-cells [17,77]. Therefore, the identification of new classes of genetic variants involved in T1D is important for better risk prediction, understanding of the pathology, and possible intervention [24,63,75,78].

In summary, genetic studies in the past found over 50 associated regions; however, most associated variants (except HLA, *PTPN22, IFIH1, CTSH, TYK2*, and *FUT2*) are involved in gene regulation and only HLA-II, *INS*, and *PTPN22* show substantial effects on T1D. Taken together, they explain fewer than 50% of the heritability in T1D, and integrative multi-omic approaches, such as genome and transcriptome analysis, epigenetics, and chromatin conformation, in relevant cell types and disease context will be important to uncover the rest of the causative variants [49,75,79]. Furthermore, T1D-linked SNPs alter the expression of many genes involved in immune signalling and β-cell destruction, as well as in viral responses. Specifically, polymorphisms in IFN-inducible genes link autoimmunity and β-cell failure with the viral response in T1D [77,80].

## 4. Environmental Risk Factors in Connection to Genetic Predisposition

Environmental effects on T1D pathology have been implied in the last century, and, in the former years, there was a plethora of different studies trying to determine its involvement in T1D. The rapid change in incidence magnitude, differences in risk of diabetes among different ethnic groups, and observations of disease prevalence in immigrants, which tend to acquire the same risk of T1D as the native population, corroborate increasing evidence for different environmental factors playing a role in T1D pathology [53,81]. Furthermore, studies of different living environments in connection with pollutants suggest their implication in T1D pathology, although the evidence for the direct involvement of air pollutants/toxins is inconclusive [82,83,84]. Instead, the risk of T1D seems to be inversely correlated to early exposure to environmental microbial diversity [85,86].

An increase in the incidence of T1D can also be associated with increased penetrance of susceptibility genes. T1D incidence peaks at pubescence, and pubertal changes may contribute to the acceleration of T1D onset in female individuals with T1D, especially in those who carry the *IL6-174CC* genotype of the *IL6* gene, regulated by oestrogen [81].

Potential triggers of islet autoimmunity include diet, toxins, and infections that affect children in utero, perinatally, and during childhood [53,87]. The birth method also appears to have a role in the predisposition to T1D since children birthed by a caesarean section are thought to have a greater risk [87,88]. All of the above can influence the perturbations in the ecosystem of the gut microbiota [89,90]. In terms of diet, cow milk, the introduction of cereals and eggs, exposure to toxic chemicals, and decreased vitamin D intake has also been proposed to increase the risk for T1D [53,81]. On the other hand, exposure to maternal microbiome, breastfeeding, vitamin D, zinc, nicotinamide, and vitamins C and E appear to protect against islet autoimmunity. Maternal microbiota and breastmilk are of special importance as they help to establish the mucosal immune system of newborns [81,91,92].

### 4.1. Viral Infections

As more epidemiological and pathological evidence builds up, infections, particularly by viruses, and their possible involvement in T1D pathology, are being investigated [93]. There is plenty of circumstantial evidence for the involvement of enteroviral infections in pancreatic and other tissues [94,95,96]. For example, individuals with T1D have detectable anti-enteroviral antibodies and Coxsackievirus B (CVB) RNA in the blood and stool 6 to 12 months before AAb development, and recently diagnosed individuals with T1D have detectable enterovirus infections within their pancreata and islets [95,97]. Moreover, in the pancreatic cellular infiltrates of individuals with T1D, there is a high expression of TLR3/4, indicative of a proinflammatory innate immune response to an infection [98]. Enterovirus capsid protein VP1 was found in the islets of individuals with T1D both close to the time of onset [80] and several years after [53], although many attempts in the past have failed to demonstrate viral existence in the pancreas because of very low viral RNA concentrations [80]. The persistence of CVB4 in pancreatic β cells disturbs insulin maturation with the release of antigenic proinsulin by β cells and causes abnormalities in cellular functions and DNA methylation, as well as a steep decline in glucose-induced insulin secretion [99]. Enteroviral β-cell tropism stems from several factors, with the most prominent being the presence of enteroviral receptors on β-cells (for example, Coxsackie and Adenovirus Receptor—CAR) [77,100]. CAR-SIV, a specific isoform of CAR, is selectively and highly expressed in β-cells [77], and CAR expression in the pancreata of individuals with T1D is increased compared to healthy controls [97,98]. Moreover, children who carried the minor SNP allele rs6517774 in the *CAR* gene region were more likely to develop islet autoimmunity [97]. Another important discovery favouring enteroviral β-cell tropism was specific β-cell intracellular factors that the virus can hijack for successful infection and replication in β-cells [77]. At the same time, β-cells are thought to upregulate a range of anti-viral proteins in response to an interferon encounter [77,101]. Furthermore, risk-associated genetic variants found in susceptible individuals are associated either with altered interferon responses [77,98,100,102], impairment of the virus clearance, or inducement of the cytokine storm and destruction of β-cells, substantiating the possible role of viruses in initiating the diabetogenic process [58]. It was also demonstrated that viral particles spread to neighbouring sites via extracellular vesicles, thus evading immune responses from the host [103]. This suggests that a prolonged low-grade infection of β-cells, owing to defective antiviral resistance, triggers the presentation of both β-cell and viral antigens at the cell surface, a potentiated inflammatory response, β-cell damage, and autoantigen release [77,100,101]. The theory is supported by observations of incomplete antibody responses to enteroviruses in children with T1D and interferon and HLA-I hyperexpression, indirect evidence for (entero)viral infection [53,80]. β-cells also display a very low expression of antioxidants and high expression of cytokine and Toll-like receptors. All this, together with the fact that β-cells are terminally differentiated, rarely proliferate, and are sensitive to the interferon response, might explain why they are so susceptible to viral infections [77].

New methods in the detection of viral RNA (for example, single-molecule in situ hybridization, smFISH) can confirm the presence of viral mRNA throughout the whole pancreas with high sensitivity, specificity, and accuracy and at lower viral loads than by classical immunostaining and even PCR [95]. In association with this is the fact that individuals with T1D are more susceptible to future infections by bacteria, viruses, fungi, and parasites, which is connected to defects of the innate and adaptive immune responses [53,104]. Another virus that is commonly associated with T1D pathology is rotavirus, although it is better documented in animal models than in humans, and the potential effects of rotavirus vaccination on T1D development remain unclear [105,106]. The world epidemic of COVID-19 indicated the possible involvement of SARS-CoV-2 in T1D pathogenesis, and there are several clinical case reports describing the new onset of T1D in COVID-19 patients [107,108,109]. Indeed, SARS-CoV-2 has been shown to infect and replicate in the cells of the human endocrine and exocrine pancreas and elicit β-cell impairment in glucose-dependent insulin secretion and β-cell apoptosis [110,111]. Whether COVID-19-induced β-cell damage is transient or permanent and whether SARS-CoV-2 can linger in the β-cell, causing chronic infection, and induce T1D on its own will require further investigations [112].

An interesting phenomenon of T1D is its seasonality, but the role of geographical location is still debatable because of sparse epidemiological data from equatorial regions, although there are reports of such seasonality disappearing close to the equator [80,81]. Seasonal patterns appear to be connected to environmental exposure to seasonal respiratory infections [81] and enteroviral infections [80] in colder months. Nevertheless, autoimmunity is probably not caused solely by a virus infection but is rather a consequence of simultaneous disadvantageous factors, for example, change in diet and sun exposure. Defects in the circadian clock regulators lead to stress in immune cells, which, in turn, exacerbate the proinflammatory response from β-cells. Again, environmental factors only play a substantial role in connection with a genetic predisposition, as is evident in children with a high-risk HLA allele, which acquired autoantibodies against islet cells during enteroviral infection more often than those without HLA risk [80]. Studies of environmental waters showed the presence of enteroviruses, and it has been suggested that free-living amoebae serve as their reservoir. Hence, this may be a manner of the transmission of these viruses and consequently contribute to the spread of T1D [113,114]. The hygiene hypothesis, particularly the lack of intestinal parasites in the more developed parts of the world, has also been linked to the spreading T1D pandemic [53,115]. There has been an increased interest in the vaccination of individuals at risk of or with overt T1D, although some evidence is not conclusive [93,116,117]. The BCG vaccine has been proposed as a possible solution in T1D prevention and treatment as it induces a response that incites the selective death of autoreactive T cells and the simultaneous expansion of Tregs. Preliminary clinical trials show promising results [53]. Another study found that AHR (aryl hydrocarbon receptor) signalling reduced autoimmune responses in T1D. AHR serves as a molecular sensor to many environmental signals, again linking their possible involvement in T1D [118].

### 4.2. Endogenized Viral Elements in the Genome

Another important group of viral entities connected to T1D pathology is human endogenized retroviruses (HERVs). They represent a potential link between genetic and environmental factors and can be transactivated by environmental viruses, such as enteroviruses, and by inflammatory stimuli [119,120]. Of note, CVB4, the enterovirus most frequently mentioned in connection to T1D pathology, was recently found to induce the transcription of a HERV-W-Env (the envelope protein of HERV-W) in primary cell cultures, such as monocytes, macrophages, and pancreatic cells [120]. Its role in T1D pathology is further supported by its detection in individuals with T1D, particularly in pancreatic acinar cells near the pancreatic lesions. Furthermore, anti-HERV-W-Env Abs have been detected in the sera of individuals with T1D and of those at risk for developing T1D, their presence overlapped with or preceded AAbs, and the extent of HERV-W-Env expression seems to be correlated with disease progression. In β-cells, HERV-W-Env inhibits insulin secretion, possibly through its interaction with TLR4, which could also lead to decreased β-cell functionality and viability. In favour of this is the fact that TLR4 signalling downstream elements, such as NF-κB, MyD88, and TRIF, are upregulated in individuals with T1D. HERV-W-Env also affects immune cells. It promotes macrophage recruitment within the pancreas, induces expression of proinflammatory cytokines in monocytes, and incites T cell responses with superantigen characteristics [119]. HERVs have also been associated with the activation of autoreactive T cells and the generation of IFN-γ [98].

### 4.3. Gut Biome

The role of the microbiome in T1D is also the subject of much interest as the presence of the specific and varied intestinal microbiota is critical in the development of the innate immune system, most importantly T_h17_ and Treg lymphocytes, for maintaining the mucosal barrier and producing different metabolites and vitamins [121]. Specifically, short-chain fatty acids (SCFAs) produced through the bacterial intestinal fermentation of dietary complex carbohydrates act as major mediators of crosstalk between the microbiome and human host. They also participate in the regulation of glucose, lipid, and energy metabolism, and modulation of gene expression, cell proliferation, and inflammation [122]. Their role in T1D is not completely understood and can be positively or negatively correlated with the risk of T1D, which depends on the microbial species and the type of SCFA produced [121,122,123]. Individuals with T1D are reported to have reduced diversity (dysbiosis) of their microbiota [53,88]. The microbiome shifts towards Gram-negative bacteria and results in increased release of LPS, thus stimulating a proinflammatory response, although most studies excluded the small intestine microbiome, which is better linked to the pancreas [53]. At the same time, intestinal permeability increases due to an SCFA-mediated impaired intestinal integrity. Accordingly, exogenous antigens and the microbial components can translocate into the circulation and promote systemic inflammation and autoimmune progression [123,124]. Consequently, alteration of the microbiome composition could lead to the loss of immune tolerance, which precedes the onset of T1D [125].

Alterations in the gut virome in children have been linked to autoimmune conditions, including T1D, although the results are inconclusive as of yet [126,127]. Studies of intestinal proteomes and metabolomes corroborated the role of microbiota in mucosal barrier function and modulating the immune response, respectively [88]. Although the mycobiome represents only 0.1% of the intestinal microbiome, its role in maintaining the homeostasis of the body seems to be significant, and the analysis of the mycobiome composition in the adults with T1D and T2D compared to the control group demonstrated differences in the profile of the gut mycobiota of individuals with T1D [128]. Studies of birth mode and breastfeeding influences on the microbiome in connection with T1D development and intestinal viromes are offering an interesting view on the matter; however, the evidence is scarce and inconclusive [88]. On the other hand, the use of broad-spectrum antibiotics early in life had a positive association with an increased risk for developing T1D [88,129]. If the intestinal microbiota influences islet autoimmunity, this should also be detectable outside the intestinal mucosa. Indeed, one group [130] examined the systemic anti-commensal Ab response against intestinal bacteria and linked high-risk HLA haplotypes with these Abs in serum and future T1D diagnosis. Diet is an important factor in intestinal microbiome composition and it, in turn, exerts its influence, not only directly stimulating an appropriate immune response but also through the epigenetic regulation of immune cells with metabolites, such as butyrate, acetate, polyphenols, and vitamins. In dysbiosis, the metabolome balance is disturbed, consequently influencing aberrant crosstalk between bacteria and the host’s immune system [131,132].

## 5. Epigenetic Factors in T1D Pathology

Epigenetic regulation is a link between genetic information and environmental influences and results in cells’ observable phenotype [133]. In T1D, it plays an important role as GWAS studies map many of the SNP variants to enhancer regions, which are important for transcription factor binding, thus regulating gene expression [49,134]. Under certain inflammatory stimuli, especially prolonged, latent enhancers form epigenetic memory through methylation marks and do not return to their latent state. Instead, they express a more potent response on the next exposure to inflammatory signals, which influences the disease onset and severity [49]. Since specific DNA methylation patterns are established during embryogenesis and foetal development through a programmed process, potential vulnerability to environmental-exposure-related epigenetic alterations at these stages may induce permanent physiological changes, potentially leading to a variety of diseases later in life. This happens through genome imprinting and environmental epigenetic modification, notably maternal enteroviral infections. Furthermore, studies have shown that prenatal exposure to maternal adverse life events results in lasting and broad functional DNA methylation changes in innate and adaptive immune genes and the genes involved in glucose metabolism [135].

SNPs can influence the regulation of chromatin conformation and the binding affinity of the transcription factors, therefore disrupting the three-dimensional genome organisation and expression of immune genes in T1D. A recent study [136] identified novel non-coding risk SNPs in the T_h1_ and Treg in individuals with T1D, which disrupted the enhancer activity by inhibiting the transcription factor binding of the genes involved in immune responses. Epigenetic modifications, specifically the methylation status of genes, are connected with T1D risk and appear to act independently of genetic variants. Hypomethylation (sign of transcription activation) of CpG sites in the *HLA-DQB1*, HLA-II connected genes, and *GAD65* gene have been observed, as well as global hypomethylation of the CpG sites within the promoter regions of the genes in individuals with T1D. The hypermethylation of the *IL2RA* gene promoter reduces the expression of the IL2RA (CD25) protein, which is important in the immune regulation of the inflammatory response by Treg cells. The methylation status of the *INS* gene promoter is considered to influence the insulin expression in β-cells and the medullary thymic epithelial cells and, therefore, have an indirect influence on the antigen presentation and maintenance of self-tolerance against β-cells [137]. Observations of longitudinally different DNA methylation patterns in individuals with T1D prior to diagnosis and islet autoimmunity, when compared to healthy controls, corroborate the idea of epigenetic modifications having an impact in T1D pathogenesis [138]. The gut microbiome interactions with the human epigenome are of special importance as bacterial metabolites, such as SCFAs, polyphenols, and vitamins, can act as cofactors for the key epigenetic enzymes. Dysbiosis affects the methylation and acetylation status of DNA and histones and influences ncRNA binding and miRNA expression in different T1D-related genes, including *NF-KB P65*, *CTLA4*, *IL2*, and *FOXP3* [31,131].

Genetic variation can also be linked to chromatin misfolding and aberrant gene expression in T cells. Single-cell transcriptional profiling of the immune cell population in the pancreases of individuals with T1D also revealed increased expression of the KRAB-Zinc finger proteins (KRAB-ZFPs) that work as repressors of specific endogenous retroviruses, again relating the involvement of these elements to disease etiopathogenesis. It will be essential to ascertain whether chromatin misfolding occurs at *KRAB-ZFP* genes in humans with T1D [139].

Taken together, the methylation status of DNA, 3D chromatin landscape, and genetic variants in non-coding regions affect cells’ transcriptome and may consequently influence the intensity of the immune response and increase the risk for T1D [49], especially in the context of in utero development [135].

## 6. Extracellular Vesicles and Non-Coding RNAs

The role of extracellular vesicles and non-coding RNAs (ncRNAs) in T1D pathology is a relatively new and exciting area of research. The evidence indicates that several groups of ncRNAs, including miRNAs and long non-coding RNAs, are partakers in regulating β-cell functions during the T1D pathological processes and potential biomarkers for early diagnosis and prognosis of the disease [140,141,142]. Recent findings by our group [143] and others [144,145,146] link extracellular vesicles (EV) from β-cells and immune cells and extracellular-vesicles-derived human-miRNAs with T1D pathology. Different subpopulations of EVs from β-cells arise from a response to inflammatory cytokines [147], act as cell-to-cell communication mediators, and are potential disease biomarkers and promising therapeutic tools [141]. They promote TLR-binding miRNA partitioning into small EVs [144], which can be transported through long distances and internalized by phagocytes, where they trigger an endosomal TLR7/8 mediated response, resulting in more cytokine production, thus mutually propelling the inflammatory response. MiRNAs presumably mimic the action of the pathogen ssRNA recognized by TLR7/8, consequently inducing the same innate response [143]. Other authors also consider the theory of molecular mimicry as being a possible cause of cross-reactivity between viral components and islet proteins. They could generate viral-human zwitter peptides (peptides that share a complete sequence homology irrespective of where they originate) [53,148]. MiRNAs have also been connected to Tregs in individuals with T1D, most notably the upregulation of miR-125a-5p and downregulation of miR-342, which could exert their influence through the obstruction of Treg migration to the pancreas and through cytokine signalling, respectively [149]. Further, miRNAs could serve as highly sensitive biomarkers, providing better information regarding the onset and severity of T1D since the changes in specific plasma miRNAs were associated with different T1D phases and established biomarkers of T1D, such as C-peptide and HbA1c [145,150]. Other ncRNAs, such as circular RNAs, regulate cellular functions and may contribute to the cytokine-mediated β-cell dysfunction occurring during the initial phases of T1D [151].

## 7. Conclusions and Perspectives

Type 1 diabetes (T1D) is a T-cell-mediated autoimmune disease of pancreatic β-cells, with the possible interplay of antiviral responses and other environmental factors on top of the genetic susceptibility. This ultimately leads to an aberrant β-cell stress response and the immune-mediated destruction of β-cells in the pancreata of predisposed individuals [11].

At the time of clinical presentation, much of the β-cell mass is usually gone, and too many children still present with severe symptoms. Furthermore, individuals with T1D are in danger of long-term complications, which is reflected in shorter life expectancy compared to other people [8]. Recent research points to T1D being a heterogeneous disease as not all the islets are affected equally and individuals with T1D respond differently to therapies [10,21]. Moreover, two distinct endotypes have been proposed recently, with different characteristics and disease progress [15]. An early hallmark of the disease is insulitis, characterized by autoreactive T cell infiltrations [12]. β-cells seem to play a pivotal role in the initiation of the disease and are not just passive targets as was traditionally believed [13,29].

Collectively, the poor response of susceptible β-cells to enteroviral exposure could lead to their persistent infection, which, combined with an aberrant INF-mediated response, could result in significant changes in β-cell functions, such as ER stress, unfolded protein response, synthesis of novel autoantigens, dedifferentiation, and apoptosis. This might lead to inadvertent hyperexpression of HLA-I, increased presentation of neoantigens, and novel epitope spreading to neighbouring cells. At the same time, chemokine-mediated APC infiltration could result in HLA-II upregulation and autoreactive naïve T cell recruitment [44,77,98,100,101,102,103]. Disease-causing variants in some genes are linked with either a diminished virus clearance or inducing an immune hyper response, suggesting that a prolonged low-grade infection of β-cells leads to potentiated inflammatory responses, which could then trigger the autoimmune process. Furthermore, gene variants in IFN-inducible genes link autoimmunity and β-cell failure with the viral response in T1D [53,80]. Research of candidate genes in recent years also points to faulty β-cell functions, further reinforcing the evidence for an active role of β-cells in T1D pathology [50,58]. On top of that, β-cells are also capable of long-distance communication and could, consequently, attract immune cells and exacerbate immune responses towards themselves [13,16,35,36,37,45]. Recent research of extracellular vesicles (EV) and EV-derived human-miRNAs further corroborates the active role of β-cells in T1D pathology. EVs with miRNA from β-cells arise from a response to inflammatory cytokines and are capable of long-distance transport. At distant locations, they trigger an endosomal TLR7/8-mediated response in phagocytes, exacerbating the inflammatory response. Moreover, miRNAs could function as sensitive biomarkers since different miRNAs are characteristic of different T1D stages [143,144,145].

The theory of environmental factors, including diet, microbiome, toxins, and, above all, infections, being responsible for the initiation of islet autoimmunity is gaining ground in recent years [53,81]. A lack of early exposure to environmental microbial diversity and the dysbiosis of the gut microbiota ecosystem are supposed to lead to the loss of immune tolerance and thus confer a greater risk for T1D development [85,86,88,125]. Enteroviral infections seem to have an important role in T1D pathology as β-cells are supposed to be susceptible to enteroviruses. There, they could establish a persistent infection, leading to an ineffective antiviral response by β-cells and subsequent attraction of immune cells [58,77,94]. Endogenized viral elements, which can be transactivated by exogenous viruses, were also found to impair β-cell insulin secretion and are associated with the activation of autoreactive T cells and the generation of IFN-γ [98,119,120].

Since epigenetic regulation links genetic information with environmental influences, it is not surprising that many GWAS studies map numerous SNP variants to enhancer regions. They can also influence the chromatin conformation and the binding of the transcription factors. Under prolonged inflammatory stimuli, latent enhancers form epigenetic memory through methylation changes. Ultimately, epigenetic regulation affects cells’ transcriptome and may consequently influence the intensity of the immune response and increase the risk for T1D [49].

Currently, only genetics is capable of detecting those at risk prior to the development of islet autoimmunity [75], and most studies are based on candidate gene association and GWAS methods [46]. Only recently, the first in-depth analysis of the potential role of genetic variants using whole-genome sequencing was performed [55]. This opened the path for the use of integrative multi-omic approaches in the future search for causative variants and new biomarkers and cell types involved in T1D [17,44,101].

The up-to-date bedside approach combines early diagnosis with early treatment and closed-loop insulin pump systems, which, in concert, help to maintain a fairly constant glucose concentration and low levels of HbA_1c_ in the long-run, thus delaying or preventing long-term complications associated with diabetes. Most preventive trials have not provided favourable long-term results and were variable in individuals’ responses, which could stem from the heterogeneity between different stages and among individuals with T1D [152]. As mentioned before, endotype theory predicts different treatments are needed for different endotypes, although all lead to similar clinical presentation [15]. Novel therapies, mediating the immune system instead of suppressing it, are already being evaluated in the clinic. Furthermore, combination therapies, targeting immune cells, cytokines, and β-cells with respect to endotype, are more likely to achieve durable remission in T1D [15,29]. There is also a strong rationale for vaccination and/or antiviral drugs against viral entities implicated in T1D pathology [153,154]. Imaging studies for assessing the β-cell mass would help to determine the disease stages and the severity of the autoimmune processes in islets. Longitudinal measurements of the pancreas volume may lead to a better prediction for risk of progression from normoglycaemia to dysglycaemia and, ultimately, to clinical onset. Moreover, determining pancreas sizes may help predict glucose variance in individuals with T1D [152]. A better understanding of the involvement of T cell autoimmunity in T1D is crucial to find early molecular markers for future detection and prevention strategies. In the future, it would be sensible to apprehend the role of environmental factors in engaging T lymphocytes and other immune cells as well as their influence on β-cells. We also need to understand how innate and adaptive immunity interact with one another and with β-cells and elucidate consecutive steps for inflammatory derailment in T1D in greater detail. In addition to this, since islet-reactive T (and B) cells have been observed in healthy people, it will be vital to understand the features of this benign state and how and why it shifts towards the pathological state. To do that, we need to form a detailed description of the cellular interplay between β-cells and the immune system and their divergence throughout the disease development according to age and disease endotypes in the peripheral blood and, more importantly, in the pancreas [15]. Accordingly, a shift towards individualized therapies based on the knowledge of the heterogeneity of the disease may improve the treatment results and, therefore, result in a better outcome for individuals with T1D in the long-run [155].

## Figures and Tables

**Figure 1 genes-13-00706-f001:**
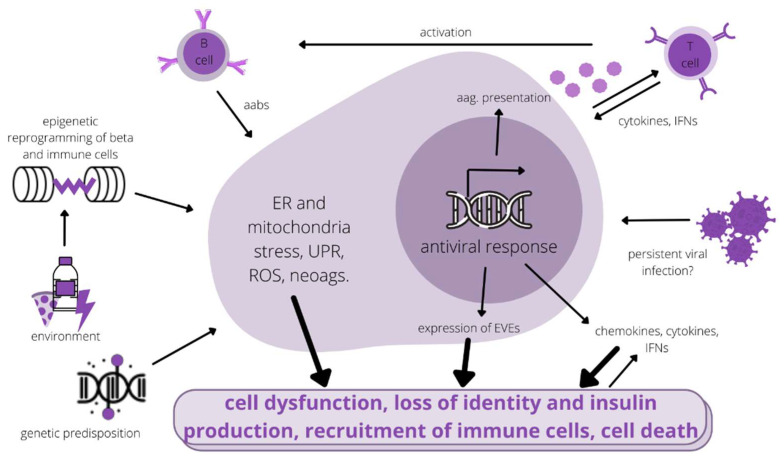
Simplified scheme of the pathogenic influences on Beta-cells in T1D. aabs—autoantibodies, aag.—autoantigen, ER—endoplasmic reticulum, EVEs—endogenous viral elements, IFNs—interferons, neoags.—neoantigens, ROS—reactive oxygen species, UPR—unfolded protein response.

**Table 1 genes-13-00706-t001:** Genes connected to T1D pathology and their influence on β-cell and/or immune cell functions.

Gene Name	Gene Function(s)	References
*BACH2*	Regulating proinflammatory cytokine-induced apoptotic pathways in pancreatic β-cells (crosstalk with PTPN2)	[46]
*C1QTNF6*	Participating in the BCR signalling pathway/cytotoxicity	[50]
*CCR5*	T_h_ cell development/chemokine-induced signalling	[50]
*CD226*	Modulating thymic T cell selection Impact on peripheral memory/effector CD8^+^ T cell activation and function Reducing regulatory functions of Foxp3^+^ Tregs	[51,52]
*CD69*	Participating in early lymphocyte activation Limiting the inflammatory response Influencing the signalling of NK cells	[53]
*CLEC16A*	Regulating the mitophagy for mitochondrial quality control Possible involvement in β-cell fragility	[54]
*COL6A6*	/	[55]
*CTLA4*	Controlling the proliferation of Tregs in the periphery Regulating pancreas autoimmunity	[56]
*CTSH*	Regulating cytokines inside β-cells for proapoptotic signal transduction	[49]
*ERBB3*	Modulating antigen presentation Modulating cytokine-induced β-cell apoptosis	[53]
*GLIS3*	Implication in the generation of β-cells, insulin expression Maintaining β-cell functions and mass Exerting antiapoptotic effects	[46,50,57]
*HIP14*	Regulating β-cell apoptosis and insulin secretion	[50]
*IFIH1*	Mediating the innate immune system’s interferon response to certain viruses Participating in β-cell response to viral dsRNA	[46,49]
*IKZF1*	Regulating immune cell development	[50]
*IL2/IL21*	Influencing T(_h_) cell differentiation and inflammatory response	[50]
*IL27*	Modulating T cell subsets and regulating inflammatory response	[53]
*IL2RA*	Variants causing abnormalities in sensitivity to IL2, which is critical to T-regulatory cell function Potential altering of the balance between Tregs and Teffs	[46]
*IL7R*	Involvement in antigen binding, Ig production, and cytotoxicity	[50]
*MRPS21-PRPF3*	/	[55]
*NRIR*	Negative regulator of interferon response	[55]
*PRKCQ*	Influencing T cell function/apoptosis/innate immune response	[50]
*PTPN2*	Inducing β-cell apoptosis after interaction with increased local levels of interferon Influencing β-cell response to viral dsRNA	[46,49]
*PTPN22*	Participating in T cell receptor signalling pathway	[53]
*SH2B3*	Participating in growth factor and cytokine signalling	[50]
*STX4*	Associated with insulin secretion Downregulating the expression of chemokine genes associated with inflammation and the apoptosis of pancreatic islets Decreasing the translocation and activation of NF-kB, thus decreasing the apoptosis	[50]
*TASP1*	Cleaving the MLL protein, which is required for proper *HOX* gene expression	[55]
*TNFAIP3*	Downregulating the intrinsic apoptotic pathway Regulating the expression levels of ZnT8 Essential for insulin production and secretion	[50]
*TYK2*	Regulating the effects of cytokines inside β-cells for proapoptotic signal transduction Mediating interferon response in connection to resistance to various infections Mediating Th1- and Th17-type immune reactions	[50,58,59]
*UBASH3A*	Downregulating the NF-kB signalling pathway upon T cell receptor stimulation, thus reducing the IL2 expression	[46,60]

## Data Availability

Not applicable.

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
