# Peer review of "Pathogenesis of Type 1 Diabetes: Established Facts and New Insights"

_genes, 2022, doi:10.3390/genes13040706_

Round 1

Reviewer 1 Report

This review article is written so nicely, covering wide range of factors including genetic risk in the pathogenesis of type 1 diabetes.

I require minor revisions as suggested below.

  • TYK2:

The manuscript listed TYK2 as one of risk genes in Table 1, describing that TYK2 to be associated with regulating the effects of cytokines inside beta-cells for proapoptotic signal transduction only. However, many papers, including citation 101, suggest that TYK2 is important to resist against diabetogenic viral infection through activation of effective interferon signaling pathway.

The authors should include this comment and had better cite recent publication, suggesting that TYK2 variant serves a risk for not only type 1 diabetes but also type 2 diabetes.

Reference; Mori, H.; Takahashi, H.; Mine, K.; Higashimoto, K.; Inoue, K.; Kojima, M.; Kuroki, S.; Eguchi, T.; Ono, Y.; Inuzuka, S.; Soejima, H.; Nagafuchi, S.; Anzai, K. TYK2 promoter variant is associated with impaired insulin secretion and lower insulin resistance in Japanese type 2 diabetes patients. Genes 2021, 12, 400, doi: 10.3390/genes12030400.

  • Important manuscript to be cited in the field of viral infection in type 1 diabetes.

The manuscript described genetic risk of coxsackie and adenovirus receptor (CXADR) gene polymorphism, correlating islet autoimmunity.

Reference: Vehik, K.; Lynch, K.F.; Wong, M.C.; Tian, X.; Ross, M.C.; Gibbs, R.A.; Ajami, N.J.; Petrosino, J.F; Revers, M.; Toppari, J.; Ziegler, A.G.; She, J-X.; Lernmark, A.; Akolkar, B.; Hagopian W.A.; Shatz, D.A; Krischer, J.P.; Hyöty, H.; Lloyd, R.E.; TEDDY Study Group. Prospective virome analyses in young children at increased genetic risk for type 1 diabetes. Nat Med 2019, 25, 1865-1872, doi: 10.1038/s41591-019-0667-0. 

Author Response

thank you for your thorough revision and suggestions and for helping us improve the manuscript.

Regarding the first issue, we have added the proposed comment regarding TYK2 and interferon signalling and the appropriate references. Aditionally, we added the proposed reference to the text, where we briefly describe shared genetic susceptibility.

Regarding the second issue, we added the proposed reference concerning the genetic risk of coxsackie and adenovirus receptor.

Reviewer 2 Report

The authors describe extensively several previously described mechanisms and recent findings in type 1 Diabetes pathogenesis.

There are minor language issues. Namely,

line 60 "This activates NF-κB activation"

line 141 "pretty low" does not sound well in scientific writing

line 285 "possible involvement of SARS-CoV-2 involvement"

line 488 "extracellular vesicles and extracellular vesicles (EV)"

Figure 1. In the figure, specifically in the nucleus, "antivral" is miswrited

Author Response

thank you for your thorough revision and suggestions and for helping us to improve the manuscript.

Regarding the first issue in the line 60: we corrected it to “This triggers NF-κB activation”.

Regarding the second issue in the line 141 (now 142): Instead »pretty low« we used “moderately low”.

Regarding the third issue in the line 285 (now 288): we deleted the second “involvement”.

Regarding the fourth issue in the line 488 (now 491): we formulated the sentence as follows –“Recent research of extracellular vesicles (EV) and EV-derived human-miRNAs further corroborates the active role of β-cells in T1D pathology.”

Regarding the fifth issue, we corrected the spelling of the word “antivral” to “antiviral” in "Figure 1".